# Viral Diseases that Affect Donkeys and Mules

**DOI:** 10.3390/ani10122203

**Published:** 2020-11-25

**Authors:** Rebeca Jéssica Falcão Câmara, Bruna Lopes Bueno, Cláudia Fideles Resende, Udeni B. R. Balasuriya, Sidnei Miyoshi Sakamoto, Jenner Karlisson Pimenta dos Reis

**Affiliations:** 1Laboratório de Retroviroses, Departamento de Medicina Veterinária Preventiva, Escola de Veterinária, Universidade Federal de Minas Gerais, Belo Horizonte 31270-901, Brazil; rebeca.jfc@gmail.com (R.J.F.C.); brulbueno18@gmail.com (B.L.B.); c.fideles91@gmail.com (C.F.R.); 2Louisiana Animal Disease Diagnostic Laboratory and Department of Pathobiological Sciences, School of Veterinary Medicine, Louisiana State University, River Rd, Room 1043, Baton Rouge, LA 70803, USA; balasuriya1@lsu.edu; 3Laboratório Multidisciplinar do Centro de Ciências Biológicas e da Saúde, Departamento de Ciências da Saúde (DCS), Universidade Federal Rural do Semi-Árido, Rio Grande do Norte 59625-900, Brazil; sakamoto@ufersa.edu.br

**Keywords:** donkeys, *Equus asinus*, mules, viral diseases, health, infectious diseases

## Abstract

**Simple Summary:**

Donkeys have been neglected and threatened by abandonment, indiscriminate slaughter, and a lack of proper sanitary management. They are often treated as “small horses.” However, donkeys and horses have significant genetic, physiological, and behavioral differences. Specific knowledge about viral infectious diseases that affect donkeys and mules is important to mitigate disease outbreaks. Thus, the purpose of this review is to provide a brief update on viral diseases of donkeys and mules and ways to prevent their spread.

**Abstract:**

Donkeys (*Equus asinus*) and mules represent approximately 50% of the entire domestic equine herd in the world and play an essential role in the lives of thousands of people, primarily in developing countries. Despite their importance, donkeys are currently a neglected and threatened species due to abandonment, indiscriminate slaughter, and a lack of proper sanitary management. Specific knowledge about infectious viral diseases that affect this group of *Equidae* is still limited. In many cases, donkeys and mules are treated like horses, with the physiological differences between these species usually not taken into account. Most infectious diseases that affect the *Equidae* family are exclusive to the family, and they have a tremendous economic impact on the equine industry. However, some viruses may cross the species barrier and affect humans, representing an imminent risk to public health. Nevertheless, even with such importance, most studies are conducted on horses (*Equus caballus*), and there is little comparative information on infection in donkeys and mules. Therefore, the objective of this article is to provide a brief update on viruses that affect donkeys and mules, thereby compromising their performance and well-being. These diseases may put them at risk of extinction in some parts of the world due to neglect and the precarious conditions they live in and may ultimately endanger other species’ health and humans.

## 1. Introduction

Horses (*Equus caballus*) and donkeys (*Equus asinus*) are mammals belonging to genus, *Equus*, which emerged approximately 4.5 million years ago [1]. Although there is an inheritance shared between them, the horses and the donkeys are notably different in their physical and behavioral characteristics [2]. The crossing of these two species results in a sterile hybrid, named mule, which has the characteristics of the two progenitor species, generating a great diversity of body types and temperaments, however, their physiological characteristics resemble those of the horse [3,4].

Currently, the world population of Equidae is estimated at 116.7 million animals, comprising 57.7 million horses, 50.4 million donkeys and 8.5 million mules [5]. Most donkey and mule populations are concentrated in the Asian continent, as well as in some countries of Central America, South America and Africa, ensuring the livelihood of 500 million people in poor communities in developing countries around the world. These donkeys and mules are used mainly as work animals (transporting cargo and traction) and for transporting people [6]. However, the industrialization and mechanization of agriculture has also led to an increasing and global abandonment of animals, which start to live in an “errant” way, roaming the roads and causing traffic accidents, in addition these stray animals representing a potential source of transmission of infectious diseases for other species, due to the lack of sanitary control [7]. Analysis of the complete genome of horses and donkeys showed divergences in genes directly involved in the inflammatory response to trauma (ITIH4) and in the regulation of cholesterol synthesis (HMGCR). These genes were positively selected in donkeys [1,8] and, they produce significant differences in physiological and biochemical parameters when compared to horses (Table 1). However, many benchmarks are still being established for specific breeds of donkeys [2,9,10].

Donkeys also have a different susceptibility to certain infectious agents and clinical manifestations when compared to horses. However, there are few scientific studies on pathogenesis, immune response and pathophysiology. Much of the available knowledge comes from the clinical experience of veterinarians, as well as knowledge, which is extrapolated from horses to donkeys [2].

Just like horses, donkeys and mules are also susceptible to infection by important virus such as equine infectious anemia (EIA), the eastern, western and venezuelan equine encephalomyelitis, Japanese encephalitis (JE), West Nile fever (WNF), equine viral arteritis (EVA), equine herpesvirus (EHV), equine influenza (EI), as well as the African horse sickness and rabies which are listed in the OIE Terrestrial Animal Health Code [13], and countries are required to report their occurrence accordingly. In addition to being relevant to the equine industry [14], some of them such as the Eastern, Western and Venezuelan equine encephalomyelitis, JE, WNF, rabies and Saint Louis encephalitis (SLE) [15,16] are of great importance for One Health. In addition, there are still unexplored viruses, such as the newly discovered *Nonprimate hepacivirus* (NPHV) [17], whose impact on equids is still poorly understood. Therefore, the main purpose of this article is to review the published literature and online resources (e.g., official donkey websites and non-governmental organizations (NGOs)) compiling and providing an update on the main viral diseases of equids on donkeys and mules.

## 2. Equine Infectious Anemia

Equine infectious anemia (EIA) is a worldwide disease caused by the *Equine infectious anemia virus* (EIAV), a *lentivirus* classified in the family *Retroviridae* that exclusively affects horses, mules, and donkeys [18]. The official EIA statistics on equids do not accurately report the prevalence of the disease in countries as they refer, and statistics are almost exclusively based on, laboratory tests carried out on the transit of animals and/or participation in events controlled by the Official Veterinary Services [19]. This most often involves animals of zootechnical interest value. It is believed that this estimate is even more compromised for mules and donkeys, which are typically animals of low value. Therefore, what is known about the occurrence of EIA in donkeys and mules comes from a limited number of studies that are summarized in Table 2.

Blood from an infected horse, donkey and mule is the most important source of EIAV for transmission to susceptible animals. The most common natural transmission of EIAV is via blood-feeding insects of the family *Tabanidae* (e.g., horse flies and deer flies) [24]. Stable flies (*Stomoxys calrcitrans*) are also capable of transmitting the virus but they are less efficient than the tabanids. Because the virus does not replicate within the insect vector, insects only serve as mechanical vectors, transferring blood though their mouth parts [25]. The virus can be transmitted iatrogenically via contaminated needles or surgical instruments, as well as transfusion of contaminated blood products. The transmission of EIAV can also occur through contact of the infected blood with exposed wounds of a healthy animal [26]. In addition to these routes of transmission, some studies have shown the possibility of spreading EIAV by vertical transmission via the uterine/perinatal route (i.e., in utero, at parturition), or following the ingestion of infected colostrum, as well as via aerosols [24,27,28,29].

The EIA clinical course is well described in horses and ponies, with little information on donkeys and mules. It is known that the presentation of clinical signs can vary according to the virulence of the infecting strain, the amount of virus to which the animal was exposed and the carrier’s immunological status [30,31]. In horses, three clinical phases of EIA have been described: acute, chronic and inapparent. The acute phase is characterized by viremia and thrombocytopenia, and occurs between 5 and 30 days after infection [18,32], they can also be accompanied by fever, lethargy, inappetence and in more severe cases may present petechiae and hemolytic anemia; however, the signs can also be mild or absent [33]. After the acute phase, the infected animal enters the chronic phase, characterized by recurrent cycles of viremia, weight loss, edema, anemia, thrombocytopenia and, less frequently, neurological clinical signs [18,33,34]. If the animal survives, the clinical signs of the chronic phase gradually decrease over a period of approximately one year [35,36]. After this period, about 90% of horses infected with EIAV evolve to the phase of inapparent carrier, where clinical signs are absent and viremia is undetectable in most cases [33,36].

Clinical, pathological, and laboratory findings of the mules show that they produce clinical signs similar to those seen in horses [35]. However, donkeys despite being susceptible to EIAV, have very low levels of viremia, which may explain the fact that they do not usually demonstrate clinical signs of the disease. Cook et al. [37] performed experimental infections of donkeys and horses with the pathogenic strain of EIAVpv strain (Pathogenic variant). All horses in the experiment showed clinical signs of EIA, while the donkeys remained asymptomatic during the 365 days of observation. Seroconversion of the animals occurred between 20- and 40-days post-infection for horses and donkeys. Interestingly, the viral load in donkeys was 100,000 times lower than in horses during the first three weeks of infection. In the case of infection with a highly virulent strain of EIAV (EIAV Wyoming), the difference in viral load was approximately 1000 times lower in donkeys than in horses, and seroconversion of the donkeys was detected two weeks after infection. In the same study, equine and donkey monocyte-derived macrophages (MDM) were equally susceptible to EIAV infection in vitro, suggesting that the clinical differences observed between these two species are not related to the permissiveness of the host cells.

A vaccine against EIA was developed in 1975 by the Harbin Veterinary Research Institute in China [38]. This vaccine was obtained through a three-step attenuation process. The first and second stages consisted of serial in vivo passage of the highly pathogenic strain EIAV_LN_ (LN40) first in horses and then in donkeys, producing the strain EIAV_DV117_ which was even more pathogenic. Thus, attenuation was obtained in the third stage, after 121 serial in vitro passages of the strain EIAV_DV117_ in primary donkey macrophages (EIAV_DLV121_). Vaccination of horses and donkeys with this cell culture adapted EIAV_DLV121_ strain provided 85% and 100% protection, respectively, when challenged with the virulent virus [39].

Due to the cost and time spent isolating leukocytes, EIAV_DLV121_ was cultured in FDD (fetal donkey dermal) cells and, after 13 passages, the new strain EIAV_FDDV13_ showed the same protective efficiency as the previous strain (EIAV_DLV121_) [39]. This viral attenuation process shows that the interaction of EIAV with horse and donkey cells is different, resulting in a distinct vaccinal protective efficacy between these species, which suggests that they may differ in susceptibility. After a previously vaccination with the attenuated strain (EIAV_DLV121_), 25% of horses and only 5% of donkeys did not develop an effective immune response to EIAV when they were challenged with the pathogenic strain (LN40) [40]. To explain the greater susceptibility of horses even after vaccination, Yin et al., 2013 demonstrated that EIAV uses the cell translation machinery of the horses more efficiently than when compared to donkeys and this would increase viral proteins expression and consequently, increase the amount of virions circulating in the animal [41]. In addition, other studies have suggested that the RNA editing enzyme equine adenosine desaminase (eADAR1) upregulates EIAV in vitro by enhancement of viral protein expression associated with the LTR region and the RRE in the env region [42]. A genomic analysis revealed multiple mutations, specially A-to-G substitutions, in the genome of EIAV_DV117_ strain, when compared to the parental strain EIAV_LN40_. The mutations were found mostly at 5′TpA and 5′ApA dinucleotides, that are commonly targeted by ADAR1, therefore ADAR1 is most likely to be related in the EIAV adaptation from horses to donkeys [43].

Even though the induction of cellular and humoral immune responses has already been demonstrated in animals vaccinated with the Chinese vaccine [44], its effectiveness is not fully proven, especially when the animals are challenged with heterologous strains of EIAV. Therefore the efficiency of the vaccine would be compromised in other regions of the world, due to the genetic diversity of strains of EIAV [45,46]. An additional disadvantage of this vaccine is the impossibility of distinguishing between vaccinated and naturally infected animals, which can compromise control measures based on serological tests [24]. With the reported limitations on the vaccine developed in China, there is no vaccine or treatment in the world for EIA and, its control occurs through the identification, segregation, and/or euthanasia of EIAV-infected animals [44,47,48,49]. The official method for diagnosing EIA in several countries around the world is the agar gel immunodiffusion (AGID) technique, also known as the Coggins Test [50]. Some countries such as the USA, Brazil, and Italy use the enzyme-linked immunosorbent assay (ELISA) as a screening method due to its high sensitivity [24,51,52,53]. There is also a Western immunoblotting test, which can be used as a third confirmatory technique, but it is not commercially available and is used only in reference laboratories in the US and Italy [29].

One of the problems of using only serological tests for the diagnosis of EIA is that there is a “immunological window”, that refers to the time after infection and before seroconversion. In horses and ponies, this period is usually less than 45 days, but periods of up to 157 days have already been reported for the AGID test [29]. Experimentally donkey seroconversion took twice as long as that for horses [37]. It is during the period of the “immunological window” that viral loads associated with blood can reach their highest levels, thus maximizing the risk of transmission [24]. There is also the hypothesis that EIAV can establish serologically silent infections similar to those seen with woodchuck hepatitis virus (WHV) and simian immunodeficiency virus (SIV) [54] that is, viruses and genetic material are identified, however by a mechanism not yet elucidated, viral replication does not stimulate detectable humoral immune response [55].

Scicluna et al. [53] observed an increase in the efficiency of the diagnosis of EIA when using three different methods in sequence, first the ELISA for screening, second the AGID to confirm the positive results in the ELISA, and third the Immunoblot (IB), to conclude on conflicting results between the two previous tests. It was observed that the frequency of samples with negative AGID and positive ELISA/immunoblot results was significantly higher in mules than in horses, which suggests a problem in the diagnosis of mules, which is important as this hybrid species has a role in the dissemination and maintenance of EIAV in the infected field herds.

Oliveira et al. [22] evaluated the seroprevalence of EIAV in a donkey population in Brazil, using the AGID, two ELISAs and the IB for discordant results. From the comparison between the tests, it was observed that the ELISA showed false-positive results, while the AGID was associated with false-negative results, which corroborates previous studies carried out in horses [29,53,56], emphasizing the need for a new diagnostic approach for EIAV research in equines, and particularly in donkeys, since they typically present an extremely weak reaction in AGID which is associated with false-negative results or results which are difficult to interpret, probably due to the delay in seroconverting or low levels of antibodies when compared to horses and ponies [37].

The other articles mentioned in Table 2 use only a single diagnostic test for EIA, the classic AGID [20,21,57] or ELISA [23]. Due to the limited techniques used in these surveys, the seropositivity found is likely to be overestimated in Sudan and underestimated in Italy, Bulgaria and Ethiopia. As reported above, although very specific, AGID is not very sensitive, increasing the number of false-negative results [53]. Therefore, the prevalence of studies that used only the AGID test is believed to be underestimated. The occurrence of EIA in Italy, Sudan, and Brazil was correlated with geographic location; regions less prone to the presence of natural fly vectors showed less seropositivity when compared to other regions within each country [20,22,23].

In circumstances where serology is not adequate, the use of the polymerase chain reaction (PCR) and even viral isolation are recommended, but these techniques have limitations. Viral isolation is extremely complex and rare, due to the difficulty, cost and time consuming for the cultivation of monocyte-derived macrophages, which prevents the routine use of this technique [24,58]. The molecular diagnosis by PCR is limited by the fact that the EIAV has a very complex global epidemiology, presenting a low level of identity between the known nucleotide sequences and their division into different clades [59], making it difficult to use universal PCR primers for its diagnosis. The effectiveness of molecular diagnosis in donkeys is still unknown, as there are no field isolates in this species that can serve as a basis for standardizing assays.

## 3. Equine Viral Arteritis

Equine viral arteritis (EVA) is a respiratory and reproductive disease that affects equides worldwide [60]. The etiological agent of EVA is the *Alphaarterivirus equid* (previously known as equine arteritis virus (EAV)), a single stranded, positive sense RNA virus that belongs to the family *Arteriviridae*, subfamily *Equarterivirinae* genus *Alphaartevirus* in the order *Nidovirales* [61].

*Alphaarterivirus equid* is transmitted via respiratory or venereal routes in horses, donkeys, and mules [11,62]. Horizontal transmission occurs via the respiratory tract after aerosolization of viral particles present in the respiratory secretions of horses with acute infection, facilitated by direct and/or close contact with susceptible animals [63,64]. Other body secretions and excretions from infected animals such as urine, aborted fetuses, and fetal membranes can also transmit the virus and play a significant role in precipitating new outbreaks of EVA [64,65]. Venereal transmission occurs exclusively via infected stallion semen in the acute or chronic phase [66].

Although *Alphaarterivirus equid* is present worldwide, there are few studies on the prevalence of this infection in donkeys and mules (Table 3). In horses, the prevalence of *Alphaarterivirus equid* varies between countries and among horses of different breeds and ages within the same country because of the management practices and genetics of host breeds that make stallions become long term *Alphaarterivirus equid* shedders [60,62]; these findings may also apply to donkeys. In the last 10 years, the presence of *Alphaarterivirus equid* in donkeys has been reported on the Asian (Eastern Anatolia, Turkey), European, and American continents (Central and South America). The highest reported prevalence was in Bulgaria (79.1%), and this study suggested that the high rate was due to a lack of control measures to prevent the disease [57]. In South Africa and Chile, *Alphaarterivirus equid* strains that infect donkeys have been reported [57]. Comparative genomic sequence and phylogenitc analyses of these strains with other equid *Alphaarterivirus equid* strains showed the existence of a new Alphaarterivirus genotypes in these regions [67,68]. In a study conducted by Bolfa et al. [69], only one donkey was positive for the presence of antibodies against *Alphaarterivirus equid*, and a possible explanation is that this animal could have been an asymptomatic carrier that could have been imported.

There are a few studies which raise the hypothesis that donkeys are more resistant to the development of the disease caused by *Alphaarterivirus equid* than horses, presenting mostly inapparent infections or mild clinical signs [73,74]. In cases where there are evident clinical signs, donkeys usually present with fever, depression, ocular and nasal discharge [11]. Similar to equine stallions, where 10–70% become inapparent carriers and maintain viral spread through semen. All the evidence indicates that donkeys are mostly asymptomatic and also act as potential reservoirs of *Alphaarterivirus equid* and help to maintain and perpetuate virus in the equid population [11,62].

## 4. Equine Hepacivirus

*Nonprimate Hepacivirus* (NPHV), also known as Equine Hepacivirus, belongs to the family *Flaviviridae* and the species *Hepacivirus A*. Currently, the NPHV is described as the homologous virus phylogenetically closest to the *hepatitis C virus* (HCV) [75,76]. The NPHV was identified in horses in 2012, and they were considered the main natural host of the virus [17]. NPHV infection was identified in serum samples collected from 1974 to 2016 in donkeys from Germany, Spain, Bulgaria, Italy, France, and Mexico as well as in mules from Bulgaria, where the presence of antibodies was detected in 31.5% (278/882) of the tested animals and viral RNA in only 0.3% (3/882), who were also seropositive, suggesting a predominance of acute infection with rapid viral depletion [77]. This could explain why studies have failed to detect the virus in donkeys and mules in Brazil (*n* = 1 and 35) [78,79], China (*n* = 14) [80], United Kingdom (*n* = 16) [81] and Italy (*n* = 134) [82], since they used only PCR-based techniques to screen the virus and also a limited sample number.

A recently published work identified that a partial NPHV E2 nuccleotide sequence isolated from an infected donkey showed a 22% divergence between the virus sequenced from horses [83]. This indicates that the virus may acquire genetic variability when it crosses the species barrier. The accumulation of variations can produce antigenically distinct variants between horses and donkeys, thereby interfering with molecular diagnosis in donkeys.

NPHV is hepatotropic, and the infection is subclinical in horses. Although the viral load remained similar between horses and donkeys, a slight increase in liver enzymes was observed in horses, whereas in donkeys, enzyme concentrations remained within the reference range [76,77]. The absence of changes in donkeys may be a reflection of the time interval between the initial infection and the tests performed since over time, horses affected by the virus tend to overcome the infection and recover to normal functioning [76]. However, it could be speculated that donkeys are more resistant to NPHV and, therefore, clear the infection more quickly, as already observed in other diseases, such as EIA.

However, it is still not clear what the impact of this new virus is on the equine industry and, to date, NPHV poses no risk to public health [84]. Meanwhile, the use of equines as an alternative animal model to studies with HCV may help to elucidate the immunological mechanisms involved in viral depletion and possibly in the natural resistance of the asinine species, which may have considerable implications for the development of effective drugs and vaccines compared to other related viruses.

## 5. Equine Herpesvirus

Equine herpesviruses (EHVs) are pathogens that establish latent infections in their hosts and affect all members of the Equidae family worldwide. They are DNA enveloped viruses belonging to the family *Herpesviridae* [85]. There are currently nine identified equine herpesviruses (EHV-1 to EHV-9), six of which belong to the subfamily *Alphaherpesvirinae* and three to the subfamily *Gamaherpesvirinae*. There exist asinine herpesvirus (AHV) of both subfamilies (Table 4) [86,87].

EHV-1 and EHV-4 are clinically, economically, and epidemiologically the most relevant pathogens within this family, causing problems in the respiratory tract of horses worldwide [97]. Both have been associated with abortion and respiratory diseases in donkeys [98]. In addition, EHV-1 is also associated with neurological diseases in donkeys [11,99,100]. Although the equine herpesvirus myeloencephalopathy (EHM) caused by EHV-1 is uncommon, it has been described in donkeys and mules in outbreaks in Ethiopia between the years 2011 and 2013. EHM was fatal in donkeys, with deaths without apparent clinical signs [101]. These viruses are antigenically and genetically related, showing cross-reactions due to similarity between 55% and 96% in the amino acid sequence of surface glycoproteins [99]. EHV-1 and EHV-4 are distributed worldwide, in domestic equine species [102,103]. However, there are few studies on the presence of antibodies against herpesvirus in donkeys (Table 5).

Some of the studies listed in Table 5 describe a higher prevalence rate of equine herpesvirus in donkeys, suggesting that they may represent an important source of infection for other equines [21,101,104,110,111]. It was also suggested that older donkeys would have a higher infection incidence rate. Depending on the study, both sexes could be considered as correlated with the occurrence of the infection. The correlation between age and disease is easily explained, as older animals have a longer time of exposure to infection; therefore, they are more likely to be naturally challenged by EHVs and recover, increasing the prevalence of antibodies against the virus [101,104,105]. Regarding the correlation between sex and disease, there were differences in results among different studies. Bolfa et al. [69] suggested that males are more susceptible to infection by EHVs 1 and 4, while Lara et al. [70] and Negussie et al. [101] suggested a possible role of females in the “maintenance” of the disease in the herd, as they are more susceptible to EHV infection.

In donkeys and mules, an infection with a gammaherpesviruses similar to *Asinine herpesvirus type 5* (AHV-5), isolated from a pharynx swab of a donkey with neurological disease in Ontario, Canada have been reported [112]. Beyond this, AHV type 4 and 5 are also considered potential causes of pulmonary fibrosis, which is a common disease in elderly donkeys [98]. AHV-5 has also been detected in donkeys with interstitial pneumonia, similar to AHV-4 [113]. *Equine herpesvirus type 7* (EHV-7), also known as *asinine herpesvirus type 2* (AHV-2), was isolated from the blood of a healthy donkey and nasal secretions of a mule after an outbreak of respiratory disease [114,115]. The first AHV-2 described in 1988 was isolated from a healthy donkey, but when inoculated into two newly weaned donkeys, it produced signs of acute rhinitis [115]. Another study showed that EHV-7 was recovered from nasal secretions in approximately 8% of healthy mules (*n* = 114) and donkeys (*n* = 13) [114]. *Asinine herpesvirus type* 3 (AHV-3) is a donkey virus that induces mild rhinitis in this species. Donkeys can also have lesions similar to the equine-coital rash, which can be caused by AHV-1 [116].

There are no specific therapeutic or biosafety procedures for AHVs. The measures that must be taken for donkeys infected with some herpesviruses must follow the protocols developed for the treatment of EHV in horses [98]. However, the unique physiology of donkeys must not be ignored. For example, respiratory infections can be aggravated because of hyperlipidemia, which is common in donkeys [11].

## 6. Flaviviral Encephalitis

The *West Nile virus* (WNV), *Saint Louis encephalitis virus* (SLEV) and *Japanese encephalitis virus* (JEV) are arboviruses belonging to the genus *Flavivirus* of the family *Flaviviridae* [117]. These viruses are transmitted by the bite of hematophagous mosquitoes and cause fatal neurological conditions in humans and horses [117]. In nature, the transmission cycle of the encephalitis, caused by those three viruses, is maintained primarily by biological vectors of genus *Culex*. Birds of the order Passeriformes may act as reservoirs and amplifiers of the WNV [118], just as water birds and pigs are potential reservoirs for JEV [119]. Humans and equines are considered accidental end-hosts and are not able to produce enough viremia to infect new vectors [117].

The neurologic clinical signs caused by these infections in horses are similar or identical to the signs induced by several other pathogens, including EHV-1 and *Rabies virus* [120]. Therefore, laboratory tests are crucial for an accurate diagnosis. Immunoenzymatic assays (i.e., ELISAs) are increasingly used because they are fast and inexpensive. However, most flaviviruses are antigenically related, which results in cross-reactivity between them, especially the SLEV, JEV, and WNV, which makes serological diagnosis more difficult [117,121]. It is often necessary to use additional confirmatory tests, especially in certain regions where there is co-circulation of those related viruses, with viral neutralization tests and plaque reduction tests (PRNT) being the most specific [119,122]. RT-PCR-based techniques can be used for direct detection of the virus and are highly specific. However, the transient nature of flavivirus viremia represents an important limitation for the use of this methodology. WNV infections, for example, have a single viremic phase of 4 to 6 days in horses. Therefore, due to the possibility of false-negative results by PCR, confirmation by serological tests is necessary, usually ELISAs for anti-viral IgM are used for this purpose [118,122].

Serological evidence of JEV infection has been reported in mules in the Himalayas and donkeys in Pakistan [123,124]. Data from the World Organization for Animal Health (OIE) show that JEV has already been identified in India, Japan, Korea, Bangladesh, Laos, Philippines, Timor-Leste, China, Papua New Guinea, and Australia [125]. However, there is no description of the affected species. JEV is a major public health problem in Asia as it is the leading cause of viral encephalitis in humans, with approximately 68,000 clinical cases per year [119]. Approximately 50% of these cases occur in China [126], where cases in equid are also frequent [125]. China is home to one of the largest populations of donkeys and mules in the world [5] but has shown a significant decline due to the deliberate culling of these species to obtain ejiao (Chinese gelatin made from donkey skin) raising the concern of experts [127]. The occurrence of infectious diseases can further aggravate this situation, as is the case of Japanese encephalitis, which has a high mortality rate, reaching up to 30–40% in more severe cases in horses, with impacts undescribed for other equids species [119].

The SLEV circulation seems to be restricted to the Americas, and serological studies in horses show its occurrence in Caribbean countries [128], Central America [129] and South America [121,130]. However, in donkeys and mules, the occurrence of the disease has only been described in Panama [129] and in two states in Brazil, Mato Grosso do Sul and Minas Gerais [131,132].

According to the OIE, the occurrence of WNV has already been reported in countries on all continents, except Antarctica, with emphasis on Israel, Hungary, Guatemala, Haiti, Canada, and the USA, which has presented cases for several consecutive years [125]. WNV infection has been described in a donkey in the south of France with neurological signs, which showed a short period of remission, followed by severe liver failure [133]. The virus has already been identified in the central nervous system of symptomatic donkeys in Brazil, who had muscle tremors, dysphagia, anterior limb ataxia, lateral decubitus, and rowing movements in the first 24 h, followed by paralysis of the pelvic limbs, loss of sensation in the spine, and mandibular trismus [134]. In the USA, three mules with clinical signs and diagnosed with WNV survived, and none of them remained in decubitus in the course of the disease, while a donkey failed to recover [135]. Antibodies have also been found in clinically healthy animals in several countries (Table 6).

Some authors suggest that donkeys and mules may be more resistant to WNV infection or may have milder clinical signs after exposure, even though severe signs may eventually occur [135,147]. Yildirim et al. [136] detected viral RNA in 28.5% (4/14) of seropositive donkeys, suggesting infection with no clinical signs. During an outbreak in Spain, WNV-positive donkeys and mules were detected in municipalities where clinical cases had not been reported, suggesting that there is greater geographical spread than previously thought and that these species can be used as sentinels for the WNV since this resistance to the virus supposedly allows the greater permanence of these animals in the herd [147].

Other studies have shown that the seroprevalence of WNV may be higher in donkeys than in horses (Table 6). Yildirim et al. [136] attributed this finding to the fact that the donkeys in the study were used mainly for the transport of water and therefore presented a greater risk of vector exposure. Davoust et al. [140] detected greater seroprevalence in donkeys than in horses, but with no statistically significant difference, while Bargoui et al. [138] did not observe a difference in the seroprevalence rate between both species, which was already expected since the animals were in similar environmental conditions and were used for the same purpose. On the other hand, some studies show a trend of greater seroprevalence in horses than in donkeys, but without statistical significance [124,143]. This is a complicated relationship to establish and a sampling discrepancy is common, and most studies do not fully explore the results obtained for donkeys and mules, while some do not differentiate between species in their data. It seems that both species have the same chance of being infected, but the rate of positivity can be influenced by factors such as geographical location, population size, and different uses of each animal species.

Neurological disorders are important health and economic threats to the equine industry. Publications from the last 20 years indicate that this morbidity represents the fifth largest cause of death among adult horses [148]. Nevertheless, little research has been carried out to evaluate the burden of neurological diseases in this species and even less in donkeys and mules, which are abundant in regions of the world that are conducive to the existence of infections due to high loads of arthropod vectors [116].

## 7. Equine Influenza

Influenza viruses makes up a large group of strains directly associated with severe respiratory infections in several species. *Influenza virus* is a member of the family *Orthomyxoviridae* and in the genus *Influenza* [149]. As the virus is airborne and is highly variable, it is constantly the target of concern related to the possible pandemics [150]. The *Influenza virus* has been classified into four types: A, B, C, and D, based on the matrix and nucleoprotein genes. Type A viruses infect animals and humans, whereas type B and C viruses infect only humans. Subtype D has been reported in pigs, cattle, sheep, and goats [151].

*Equine influenza virus* (EIV), classified as type A, is considered one of the most important viral respiratory pathogens, causing a disease with high morbidity. Equine influenza disease is mainly caused by the virus subtypes, H7N7, first recognized in 1956 [152] and H3N8, isolated for the first time in 1963 [153]. Since then, the increase in the transit of infected animals without clinical signs and which are subjected to inadequate quarantine procedures, has caused EIV to spread worldwide, except in a small number of island countries, including New Zealand and Iceland [154].

Once in the host, EIV has a tropism for the ciliated cells of the upper and lower respiratory tract, inducing necrosis of the epithelial cells, exudation of fluids rich in proteins, and agglomeration of the cilia [155]. The onset of clinical signs occurs between 5 and 14 days after infection. In horses, they are characterized by dry cough, fever, lethargy, anorexia, enlarged lymph nodes, tachycardia, hyperemia of the airways and conjunctival mucous membranes, serous nasal discharge, edema of the limbs, pain, muscle stiffness, and abortions [156]. Eventually, infected animals may develop more severe clinical signs, characterized by myocarditis and chronic obstructive pulmonary disease. Despite the high morbidity, the mortality rate is low and usually occurs in cases of pneumonia with sequelae [157].

Donkeys and mules, once affected, develop clinical manifestations similar to horses; however, in donkeys, the symptoms are more clinically severe with signs of typical broncho-interstitial pneumonia characterized by necrotizing bronchiolitis, hemorrhages, the presence of extremely swollen alveoli, and fibrinous exudation are commonly observed [158]. There is evidence from outbreaks in China between 1993 and 1994, that while mules and horses display mild clinical signs, donkeys infected under the same conditions of exposure to the virus were severely affected, and most of them died [149,159]. The greater susceptibility of donkeys to EIV infection is due to the greater propensity of these animals to develop bacterial bronchopneumonia [160]. Nevertheless, Rose et al. [161] attributed the greater severity of clinical signs seen in these animals, with the recurrent coinfection with pulmonary nematodes *Dictyocaulus arnfieldii.*

Although, approximately 50.5% of the world equine population corresponds to donkeys and mules [5], so far, infection by EIV in these animals is considered rare, even with the greater predisposition of donkeys to develop the severe form of the disease when infected. Epidemiological data regarding the presence of EIV in donkeys are limited to outbreaks of respiratory disease, caused by the H3N8 sub-lineage in Xinjuang in 2007, which affected approximately 13,600 animals [162], as well as in Shandong in 2017, limited to one property, and 300 seropositive animals were affected with a 25% mortality [158]. More recently, in 2020, in the city of Liaocheng, 120 unvaccinated donkeys showed an average seropositivity of 32.5% [163]. All of these outbreaks have been described in China’s provinces, indicating that the EIV is a major threat to the country’s large donkey farms.

Currently, vaccination is the most effective strategy, along with isolation measures, animal traffic control, and the adoption of basic biosafety measures to prevent EIV infections or limit their consequences [164,165]. However, as they are not mandatory, vaccination against EIV in herds, mainly of donkeys and mules, which are neglected species, is uncommon. However, we believe that the continuous circulation of EIV emphasizes the need for effective surveillance in herds of equines, which includes maintaining the updated vaccination schedule and applying diagnostic tests to detect subclinical cases before the animals move between different locations.

General information (taxonomy, natural host, transmission, diagnostic method and prevention) of the aforementioned viruses is summarized in Table 7.

## 8. Other Viral Diseases That Affect Donkeys

Currently, there are several diseases affecting members of the Equidae family worldwide considered of compulsory notification by the OIE; however only eleven are caused by viruses, and these include the equine infectious anemia, the equine encephalomyelitis caused by flaviviruses or alphaviruses, the equine influenza, infection with EHV-1 and *Alphaarterivirus equid*, as well as the African horse sickness and rabies [168]. Although several studies have demonstrated that donkeys and mules as well as horses are also affected by these diseases, almost all experimental studies, including epidemiological surveys, are carried out on horses, and there is little information about the pathogenesis of these diseases in donkeys and mules.

## 9. Rabies

Rabies is a zoonotic disease caused by *Rabies virus* (RABV), a highly neurotropic *Lyssavirus*, which belongs to the family *Rhabdoviridae*, order *Mononegavirales*, whose main transmission mechanism is through the bite of infected carnivorous animals (e.g., dogs, jackals, hyenas, and foxes) or blood-sucking bats [11].

Equidae are the animals most susceptible to RABV [169] and the first clinical signs usually appear between 2 and 9 weeks after infection and are initially characterized by fever, mild lameness, and cramps. However, with active replication of the virus, Equidae when infected can develop three distinct forms of rabies: the mute form characterized by muscle tremors, excessive salivation, ataxias, and depression [170], the paralytic form with ascending paralysis, loss of tone of the tail and anal sphincter [171] and the furious form, characterized by hydrophobia, photophobia, hypersensitivity to touch, difficulty in swallowing, aggressiveness, and self-mutilation [172]. In mules and donkeys, clinical signs such as abnormal vocalization, hyperesthesia, colic, self-bites, tendency to bite other animals, restlessness, and excessive salivation, are commonly observed, followed by a flaccid tail, phimosis, complete decubitus, and death [173].

Although some studies demonstrate that mainly working donkeys are an important source of transmission of RABV to humans, due to recurrent accidents involving deep wounds caused by bites which result in large scale loss of tissue, especially in children [174,175,176], the occurrence of rabies in donkeys is still limited to isolated cases in Canada, China and some countries in Africa and the Middle East [173,177,178,179,180].

## 10. Alphaviral Encephalitis

The *Eastern equine encephalitis virus* (EEEV), *Western equine encephalitis virus* (WEEV), and *Venezuelan equine encephalitis virus* (VEEV) are enveloped, single-stranded, positive-sense RNA viruses that belong to the genus *Alphavirus* in the family *Togaviridae*, that cause neurological disease in equids (horses, mules, donkeys and zebras) and humans [181]. The EEEV and WEEV have been reported in several countries in the Americas, while VEE seems to be limited to South and Central Americas [182,183]. They are transmitted by mosquitoes, as biological vectors. Passerine birds (EEEV and WEEE) and rodents (VEEV) are the main reservoir hosts, with high viremia to infect vectors, while equids and humans are considered dead-end hosts, except for VEEV, where equids are efficient amplificators of the virus and can act as a source of infection for mosquito transmission [117,182].

These viruses affect the nervous system, so the clinical signs can be identical for EEV, VEEV and WEEV, with severe neurologic disease and frequently death [120]. The mortality rate of EEEV and VEEV infection can reach up to 90% in most of the equine cases, while WEEV is least lethal in horses, with a mortality rate of approximately 30% [184,185]. Donkeys and mules exhibit similar susceptibility and clinical signals to horses [116,186]. In 1943, cases of fatal VEEV had occurred among horses, asinines and mules in Trinidad. A suspension of donkey brain tissue was prepared and injected intra-cerebrally into guinea pigs, originating the virulent Trinidad Donkey strain, from which was developed the attenuated vaccinal TC-83 strain [117,186]. EEEV infection in donkeys and mules was reported in Brazil, with a fatal case in a donkey [187]. There has been a dramatic reduction in cases of WEEV during recent decades, with no outbreaks between equids reported since 1999 [183]. Even for EEEV and VEEV, reports on donkeys and mules are scarce. The neglect of these species can have negative impacts, especially regarding VEEV, since they also are important amplifiers of the virus. For this reason, donkeys and mules should be included in vaccination programs. This is more relevant in countries such as Brazil, which has one of the biggest populations of feral donkeys in the world [5].

## 11. African Horse Sickness

African horse sickness (AHS) is a disease caused by the *African horse sickness virus* (AHSV), an *Orbivirus* belonging to the *Reoviridae* family, non-contagious, transmitted by mosquitoes of the genus *Culicoides*. AHSV has a predilection for vascular endothelial cells, and equines when infected can develop up to four distinct forms of the disease: the cardiac form characterized mainly by edema, ocular and tongue hemorrhages, the pulmonary form where the infected animal has difficulty breathing, dyspnea, cough, and sweating, the mixed form that occurs in a combination of the cardiac and pulmonary forms and the form named sick fever characterized by moderate fever and some edema of the supraorbital fossae [188]. Although the risk of mortality in affected horses may reach 95% [189,190], experimental infections carried out on African donkeys have shown the presence of only minimal and mild histopathological lesions characterized by the accumulation of fluid in the peritoneal cavity, petechiae, and equimotic hemorrhages in the left hepatic ligament. Due to this, it is widely accepted that feral donkeys and mules are resistant and most of these animals, when affected, become asymptomatic carriers of the virus [191].

Currently, AHS is a disease restricted to sub-Saharan Africa and epidemiological data from Zimbabwe and Ethiopia, both African countries, have a prevalence ranging from 59.3% to 75% in donkeys as well as 55.5% in mules [192,193]. Although typically limited to North Africa and Middle Eastern countries, epizootic events affected Spain in 1987, after the importation of several infected zebras with no apparent clinical signs. However, in 1988, the disease reemerged in the country and killed about 13,000 horses. Studies have indicated that viral reemergence in Spain occurred due to the clinical resistance of mules and donkeys with no apparent clinical signs and inaccurate detection which facilitated disease spread in the region [194]. Recently, a new outbreak of AHS was reported in different provinces of Thailand, infecting about 422 animals and leading to 386 deaths. Although AHS is not a transmissible disease, possibly the number of infected animals is still increasing, due to the large population of insect vectors in that region [195], representing a significant threat to other Asian countries, including China, which has the largest equids population in the world [196] and about 6.8 million animals without a history of AHS vaccination [197].

Although the presence of infections caused by RABV and AHSV are compulsory notifications, the low economic importance of donkeys and mules, when compared to horses, make studies on the pathogenesis of diseases that affect the Equidae family, increasingly scarce for these species and, consequently, the world population of 50.4 million donkeys and 8.5 million mules poses a significant risk, acting as a reservoir for the maintenance and transmission of important pathogenic viruses. Considering their unique physiology and their natural resistance to the development of clinical signs when affected by some diseases, it would be important for donkeys and mules to have the same focus of interest when it comes to infection since they are abundant in number and have a close coexistence with horses in various regions of the world.

## 12. Conclusions

Most of the viral diseases affecting the family *Equidae* are considered compulsory to be reported to the OIE [168]. In the absence of an effective vaccine, the control of an infectious disease is dependent on breaking the transmission cycle through the use of highly sensitive and specific diagnostic methods capable of detecting positive animals, followed by segregation of them from the rest of the herd, and reporting to state and/or government authorities.

Donkeys and mules are susceptible to infection by viruses such as EIAV, *Alphaarterivirus equid*, NPHV, EHVs, WNV, SLEV, JEV, EIV, RABV, EEEV, WEEV, VEEV, and AHS. However, in many cases, when infected, they show resistance/low susceptibility to the development of clinical signs. In this situation, infected animals are hardly identified and remain for long periods as possible sources of infection for horses and other species, including humans. In addition, with regard to the natural “resistance” of donkeys and mules linked to their growing devaluation, there is a significant deficit of studies related to the epidemiology and pathogenesis of viral infectious diseases in these animals. As a result, the control of infectious diseases in donkeys and mules is most often compromised. Furthermore, it would be important to create robust funding for studies of infectious diseases in donkeys and mules with the development of diagnostic tests specific to them, when necessary, and that the OIE and regional agencies in countries where donkeys and mules are abundant adopt preventive measures against the viral spread in these animals, with compulsory disease notification in these populations.

## Figures and Tables

**Table 1 animals-10-02203-t001:** Comparison of the physiological parameters between donkeys and horses. Source [11,12].

Physiological Parameters	Donkeys	Horses
Temperature (°C)	36.5–37.8	37.5–38.5
Pulse (Beats/Minute)	36–52	30–40
Respiration (Breaths/Minute)	12–28	18–20
**Hematology and Biochemistry Parameters**		
RBC Count (10^6^/µL)	4.4–7.1	6.0–10
Hemoglobin (g/dL)	8.9–14.7	12–17
MCV (fl)	53–67	42–58
Triglyceride (mmol/L)	0.6–2.8	0.05–0.5
Total Bilirubin (µmol/L)	0.1–3.7	0.06–0.12

**Table 2 animals-10-02203-t002:** Studies on the occurrence of equine infectious anemia (EIA) in horses, donkeys, mules and zebras from 2009 to 2019.

Disease	Species	% Positivity	Location	References
EIA	Donkey	0% (0/1568)	Italy	[20]
	Horse	0.2% (41/23971)		
	Mule	3.5% (27/767)		
	Zebra	0% (0/3)		
	Donkey	0.2% (1/662)	Ethiopia	[21]
	Horse	0% (0/289)		
	Mule	0% (0/51)		
	Donkey	3.27% (12/367)	Brazil	[22]
	Donkey	8.28% (14/169)	Sudan	[23]
	Horse	3.17% (6/189)		

**Table 3 animals-10-02203-t003:** Studies of the occurrence of *Alphaarterivirus equid*, including donkeys and/or mules, from 2009 to 2019.

Disease	Species	% Positivity	Location	References
EVA	Donkey	79.1% (152/192)	Bulgaria	[57]
	Donkey	20% (17/85)	Brazil	[70]
	Donkey	2 animals	Chile	[68]
	Donkey	2.5% (1/40)	West Indies	[69]
	Donkey	8.3% (19/227)	Turkey	[71]
	Horse	15% (29/193)		
	Donkey	3.46% (53/1532)	Turkey	[72]

**Table 4 animals-10-02203-t004:** A list of *Herpesviruses* that infect the family *Equidae.*

Virus	Synonym	Subfamily	Genus	Size of Genome (kpb)	Percent Identity	Natural Host	Disease
EHV-1	*Equine abortion virus*	*α*	*Varicellosvirus*	150	EHV-4	*Equus caballus*	Respiratory, Abortion, Neurological
					55 to 96%		
EHV-2	Old *equine eytomegalovirus*	*γ*	*Percavirus*	184	EHV-5	*Equus caballus*	Rhinitis and Conjunctivitis
					60%		
EHV-3	*Coital exanthema virus*	*α*	*Varicellosvirus*	151	EHV-1/EHV-4/EHV-8/EHV-9	*Equus caballus*	Coital Exanthema
					63%		
EHV-4	*Equine rhinopneumonitis virus*	*α*	*Varicellosvirus*	146	EHV-1	*Equus caballus*	Respiratory
					55 to 96%		
EHV-5	Old *equine cytomegalovirus*	*γ*	*Percavirus*	179	EHV-2	*Equus caballus*	Equine Multinodular Pulmonary Fibrosis
					60%		
EHV-6	*Asinine herpesvirus* 1	*α*	*Varicellosvirus*			*Equus asinus*	Coital Exanthema
EHV-7	*Asinine herpesvirus* 2	*γ*	*Rhadinovirus **			*Equus asinus*	Undefined. Virus has been found in nasal secretion of donkeys and mules
EHV-8	*Asinine herpesvirus* 3	*α*	*Varicellosvirus*	149	EHV-1/EHV-9	*Equus asinus*	Rhinitis
					89%/92%		
EHV-9	*Gazelle herpesvirus*	*α*	*Varicellosvirus*	148	EHV-1	*Equus grevyi*	Neurological
					86 to 95%		
AHV-4	*Asinine herpesvirus* 4	*γ*	Unclassified			*Equus asinus*	Pneumonia
AHV-5	*Asinine herpesvirus* 5	*γ*	Unclassified			*Equus asinus*	Pneumonia, Equine Multinodular Pulmonary Fibrosis
AHV-6	*Asinine herpesvirus* 6	*γ*	Unclassified			*Equus asinus*	-
ZHV	*Zebra herpesvirus*	*γ*	Unclassified			*Equus zebra*	Pneumonia
WAHV	*Wildass herpesvirus*	*γ*	Unclassified			*Equus somalicus*	-

*α*: Alphaherpesvirinae; *γ*: Gammaherpesvirinae. * It may be a member of the genus but has not yet been approved as a species. Source [88,89,90,91,92,93,94,95,96].

**Table 5 animals-10-02203-t005:** Studies on the occurrence of equine herpesvirus, including donkeys and/or mules, from 2009 to 2019.

Disease	Species	Agent and % Positivity	Location	References
EHV	Donkey	EHV-1/24.2% (31/128)	Turkey	[104]
	Horse	EHV-1/14.5% (42/290)		
	Mule	EHV-1/37.2% (32/86)		
	Donkey	EHV-1 and 4/0% (0/4)	Brazil	[105]
	Horse	EHV-1 and 4/18.7% (140/749)		
	Mule	EHV-1 and 4/6.8% (5/73)		
	Donkey	EHV-4/69.7% (134/192)	Bulgaria	[57]
	Donkey	EHV-1/20.2% (21/104)	Ethiopia	[21]
		EHV-4/84.6% (88/104)		
	Horse	EHV-1/7% (7/100)		
		EHV-4/91% (91/100)		
	Mule	EHV-1/0% (0/4)		
		EHV-4 100% (4/4)		
	Donkey	EHV-1/33.3% (4/12)	Tanzania and Namibia	[106]
		EHV-9/0% (0/12)		
	Somali wild ass	EHV-1/73.6% (14/19)		
		EHV-9/5.2% (1/19)		
	Zebra	EHV-1/61.8% (55/89)		
		EHV-9/29.2% (26/89)		
	Donkey	EHV-1/2 animals	EUA	[107]
	Horse	EHV-1/115 animals		
	Mule	EHV-1/2 animals		
	Donkey	EHV-1/51.85% (126/243)	Turkey	[108]
		EHV-4/64.2% (156/243)		
	Horse	EHV-1/52.48% (222/423)		
		EHV-4/83.69% (354/423)		
	Donkey	EHV-1 and 4/69.5% (57/82)	Sudan	[109]
	Horse	EHV-1 and 4/65.9% (83/126)		
	Donkey	EHV-4/14.8% (16/108)	Iran	[110]
	Mule			
	Donkey	EHV-1/82 animals	Ethiopia	[101]
	Horse	EHV-1/6 animals		
	Mule	EHV-1/3 animals		
	Donkey	EHV-1 and 4/74.7% (201/269)	Ethiopia	[111]
	Horse	EHV-1 and 4/66.7% (64/96)		
	Mule	EHV-1 and 4/50% (6/12)		
	Donkey	EHV-1/10% (4/40)	West Indies	[69]
		EHV-4/53% (21/40)		
	Horse	EHV-1/27.6% (39/140)		
		EHV-4/90% (126/140)		
	Donkey	47% (40/85)	Brazil	[70]

**Table 6 animals-10-02203-t006:** *West Nile virus* (WNV) seroprevalence studies that include donkeys and/or mules.

Disease	Species	% Positivity	Location	References
WNV	Donkey	20% (14/70)	Anatolian Province (Turkey)	[136]
	Horse	0.8% (1/118)		
	Donkey	1.28% (3/234)	Turkey	[137]
	Horse	4.15% (27/650)		
	Equids ^1^	28% (332/1189)	Tunisia	[138]
	Donkey	8.6% (5/58)	Borno State (Nigeria)	[139]
	Horse	11.5% (11/96)		
	Donkey Horse	55.4% (249/449)	Punjab and Khyber Pakhtunkhwa Provinces (Pakistan)	[124]
	Donkey	86.2% (25/29)	Northwest Senegal	[140]
	Horse	68.7% (44/64)		
	Donkey	14.4% (33/222)	Northeast Algeria	[141]
	Horse	26.8% (19/71)		
	Donkey	39.3% (50/127)	Palestine and Israel	[142]
	Horse	82.6% (380/460)		
	Donkey	15% (6/40)	Leeward Islands (West Indies)	[69]
	Horse	18.6% (26/140)		
	Donkey	12.7% (83/150)	Northern Egypt	[143]
	Horse	20.7% (83/400)		
	Donkey	47.6% (10/21)	Guadaloupe Archipelago	[144]
	Horse/Pony	22.3% (69/309)		
	Donkey/Mule Horse	25% (4/16)	Chiapas and Puebla States (Mexico)	[145]
	Equids ^1^	1.8% (4/217)	Mato Grosso State (Brazil)	[146]

^1^ Authors did not specify the evaluated species.

**Table 7 animals-10-02203-t007:** Summary of equine viral infectious diseases, listing the infectious agents, their taxonomic grouping, natural hosts, clinical characterization, transmission mechanisms, diagnosis methods and the precautions required to prevent their spread, as determined by World Organisation for Animal Health (OIE) terrestrial manual.

Virus	Synonym	Family	Genus	Genome		Natural Host	Disease	Transmission	Diagnostic Method	Prevention
				Type	Size (kbp)					
EIAV	*Equine infectious anemia virus*	*Retroviridae*	*Lentivirus*	ssRNA+	8.2	*Equus caballus* *Equus asinus*	Acute and Chronic Disease	Inoculation of contaminated blood by blood-feeding vectors and fomites	AGID, ELISA and IB *	Application of effective biosecurity measures and vector control
EAV	*Equine arteritis virus*	*Arteriviridae*	*Alphaartevirus*	ssRNA+	~12.7	*Equus caballus* *Equus asinus*	Respiratory and Reprodutive Disordrs	Respiratory or veneral routes	VI, RT-PCR, RT-qPCR, immunohistochemistry, or serological assays (ELISA, PRNT)	Application of effective biosecurity measures and vaccination
NPHV	*Nonprimate hepacivirus* or *Equine hepacivirus*	*Flaviviridae*	*Hepacivirus*	ssRNA+	9.5	*Equus caballus*	Subclinical Hepatitis	Blood/serum transfusion and possibly by vertical route	Unstandardized	Test of blood/sérum before transfusion
WNV	*West Nile virus*	*Flaviviridae*	*Flavivirus*	ssRNA+	~11.0	Aquatic and passeriformes birds	Neurological Disease and Possibly Death	Bite of Culex spp.	PRNT, PCR and/or ELISA-IgM	Vector control and vaccination (available for horses)
SLEV	*Saint Louis encephalitis virus*					Birds	Neurological signs can be mild or severe. Death rarely ocorrs.			Vector control. Vaccine is not available
JEV	*Japanese encephalitis virus*					Pigs and quatic birds	Neurological disease and possibly death			Vector control and vaccination (available for horses, swin and human)
IV	*Influenza virus*	*Orthomyxoviridae*	*Influenza*	ssRNA-	8 ssRNA- segments **	*Equus caballus Equus asinus*	Respiratory disese	Respiratory routes and fomites	Clinical diagnosis, VI, influenza A antigen detection, haemagglutinin inhibition, and qPCR	Application of effective biosecurity measures and vaccination

AGID: agar gel immunodiffusion; ELISA: enzyme-linked immunosorbent assay; IB: Western immunoblotting test; VI: virus isolation; PCR: polymerase chain reaction; RT-PCR: reverse transcription–PCR; RT-qPCR: real-time RT-PCR; qPCR: real-time quantitative PCR; PRNT: plaque reduction neutralization test. * Confirmatory technique, but it is not commercially available. ** EIV virion consists of eight single stranded negative sense RNA segments ranging from 866 to 2314 pb. Source: [24,62,117,118,119,155,166,167].

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
