# Peer review of "Viral Diseases that Affect Donkeys and Mules"

_animals, 2020, doi:10.3390/ani10122203_

Round 1

Reviewer 1 Report

Reviewer Report 1: This is a very comprehensive review article that is centered on viral diseases which affect donkeys and mules welfare. These animal species are facing serious extinction from humans and certain viral diseases that the authors emphasized on.

ABSTRACT:

Line 23: correct put 50 % and not 50%

Lines 34 and 35: Recast as: Therefore, the objective of this article is to provide a brief update on viruses that affect donkeys and mules, thereby compromising their performance and well-being.

Line 38: Viral diseases

INTRODUCTION

Lines 57, 68 and 70: correctly put 50 %, 40 % and 28 %, respectively

Line 97: Mostly and not mostley

Equine Infectious Anemia

Line 114: Stable flies (Stomoxys calcitrans)- correct the spelling for the sp.

Lines 115, 116 and 117: Recast as: The virus does not replicate within the insect vector, insects only serve as mechanical vectors, transferring blood through their mouth parts.

Lines 134, 156, and 163: Correctly put 90 %, 85 and 100 %, 25 and 5 %, respectively

Line 168: Ref [39] repeated twice,-delete one

Line 170: italicize in vitro to be in vitro

Equine Viral Arteritis

Line 227: Previously known as and not ad

Line 254: Recast as: There are a few existing studies which raise the ----

Line 258: Correctly put where 10 - 70 % become ----

Equine Hepacivirus

Lines 267 and 268: Correctly put 31.5 % and 0.3 %, respectively

Line 274: Correctly put 22 %

Equine Herpesvirus

Line 304: Correctly put 55 % and 96 %, respectively

Line 319: Table 4. A list of Herpesviruses that infect the family Equidae - check the spelling of Herpesviruses, letter s is omitted

Line 330: Correctly put 8 %

Line 338: Table 5, under Agent and % positivity, give a space between the values and the percentages. When you do this, the table will be much clearer

Flaviviral Encephalitis

Line 341: Write WNV, SLEV, and JEV in full before putting the abbreviations in brackets because this is the first time they are appearing in the text,e.g. West Nile Virus (WNV), etc.

Lines 369 and 373: correctly put 50 % and 30 - 40 %, respectively

Line 390: Table 6, under % positivity, give a space between the values and the percentages. When you do this, the table will be much clearer

Line 394: Correctly put 28.5 %

Equine Influenza

Lines 457, 463 and 464: Correctly put  50.5 %,  25 % and 32.5 %, respectively

Alphaviral Encephalitis

Lines 516 and 517: Correctly put 90 % and 30 %, respectively

African Horse Sickness

Lines 538, 544 and 545: Correctly put 95 %, 59.3 % to 75 % and 55.5 %, respectively

Conclusions

Lines 578 and 579: Recast as: At this place the animals stay for days or weeks in precarious conditions leading to the occurrence of infectious diseases that pose a risk to the animals at the quarantine facility.

Line 583: Delete 'warns of' and replace with 'calls for'. Also, it calls for the need to develop new ----

References

Line 666, ref 29: Hyperglobulinemia, letter n is omitted

Lines 941 and 942, ref 126: West Nile Virus, check the spelling

Line 1052, ref 171: Not properly cited

Line 1081, ref 183: The word "Available" is repeated twice, delete one

Reviewer 2 Report

In this review manuscript, Falcão Câmara et al. provide a broad review of viral diseases that affect donkeys and mules, with comparisons to knowledge about the available data for viral diseases of horses.  The review effectively highlights the importance of the subject, the paucity of published data on the subject, and identifies knowledge gaps that should be addressed by the research community.  In many cases, the data are presented along with factors to consider that may skew the overall results (e.g. small sample size, economic factors that affect resources available for studying the subject).  For the most part, the text is clear, although now and then it veers away from the subject of infectious diseases, which distracts from the focus of the review.  Detailed comments are listed below.

1) The manuscript addresses the issue of the the ejial trade (gelatin made from donkey skin) in China; however, the information in the introduction and in the conclusion are not coherent.  In the introduction, the reader is presented with what appears to the be the large-scale trade of live animals that are exported from South America to China.  But in the conclusion, the authors instead describe large numbers of animals being slaughtered in South America, and the harvested tissue being exported.  Firstly, the authors should consolidate the discussion of this practice in the introduction.  Secondly, the authors should clarify which of the two situations they describe accurately reflects the trade.  Thirdly, following a clear description of the situation (with references), the authors should then explain the impact on donkeys and the possible spread of infectious diseases—the topic of the review.

2) On line 97, there should be a reference added for the NPHV virus.

3) On line 102, it is unclear what is meant by the term “cosmopolitain disease”, a different term should be used.

4)  On line 514, it is unclear what viruses the authors are referring to when they state that “so the clinical signs can be identical for both,”.  The text should be modified to make this clear.

5) On line 577, the authors must provide a reference for the statement that “At this place, the animals stay for days or weeks in precarious conditions leading to the occurrence of infectious diseases that pose a risk to them itself in addition to….”

6)  Several references to hyperlinks lead to dead links, namely:

# 68:  ICTV Genus: Hepacivirus…

#79:   ICTV  Herpesviridae

7)  Reference #94, King et al. appears to be incomplete, the volume and page numbers are absent.

8) Reference #168, Spickler, appears to be incomplete.

9)  The conclusion should be completely rewritten to focus on the main subject of the review, namely viral diseases of donkeys and mules.  Presently, the conclusion is a critique of the ejial trade, and does not relate to the subject of the review.

Minor points

Line 57:  Replace “reduced” by “dropped”.

Line 88:  Remove “Source [12]”.

Line 105:  A coma (i.e.[ ,]) should be added after “based on”.

Line 114:  The spelling of the term Stomoxys calcitrans” should be corrected.

Line 114:  The word “are” should be added after “(stomoxys calcitrans)”.

Line 145:  The word “smaller” should be replaced by “lower”.

Line 168:  The citation [39] appears to be in duplicate.

Line 254:  It appears that the word “that” is missing after the word “studies”.

Line 296:  The authors likely meant to say “There exist asinine herpesviruses of both subfamilies…”

Table 4:  The typical nomenclature for herpesviruses uses the lowercase Greek letters a and g.

Table 4:  The official name of the genus is “Varicellovirus”.

Line 334:  The word “some” should be replaced by “a”.

Line 433:  The word “developing” should be replace by “causing”.

Reviewer 3 Report

This review summarizes the most important viral diseases that affect Equus genus. The authors offer information about clinical signs and symptoms, transmission, detection methods as well as treatment and prevention. The authors describe viral diseases have veterinarian importance and zoonotic potential. The main problem with this review is that lack of consistency in the presentation of the study and a loss of focus in the introduction. Al the issues listed below should be improved in order to made this review suitable for publication.

Major points:

1.Scientific nomenclature and taxonomy are incorrectly used on many occasions: Line 41, Equus genus, genus name must be write always capitalized and italicized or underlined. Line 92, Herpesviruses refers to a group of viruses from Herpesviridae or Herpesvirus, so Herpesviruses as a group should not be highlighted in italics. Line 190 WHV and SIV are not correctly write. This problem is common in all the text.

2.Sometimes there are statements or data that lack a scientific citation, for example: line 92 “Agents such as Equine infectious anemia virus (EIAV)…”, line 113 “The most common natural transmission of EIAV…”, line 156 results of vaccination…. This error is made several times in the writing and must be corrected throughout the whole article.

Viral strains causing EIA are no written in the same way. For example, line 155 EIAVDLV121 and line 156 EIAVDLV121.

3.The tables used to describe the diseases are not the same in content for each virus. The same format and content should be included for each disease. For example, in some cases the symptoms are included in text and in others in tables. Table #4 is a good model to use in the other diseases. It is also recommended for a better understanding of the reader, organize, and summarize in tables contents such as: transmission, diagnostic method, treatment, and prevention.

4.For EIA vaccination authors describe 85 and 100% of protection for horses and donkeys, respectively. However, in line 175 is affirmed that “There is no treatment or effective vaccine for EIA”. Is LN40 pathogenic strain the most common causal agent of EIA?. What is the difference between the protection results indicated in line156 versus line163?. Was the second study conducted with strain  117 or LN40?

5.Reference #39 are twice and different written in the text in line 168.

6.The Influenza virus particle and genome description (line 424-431) is not necessary or relevant for the text. Also, this information is not included for other virus mentioned in the review. This type of content must include for all causal agents or eliminated, so that the information provided is homogeneous.

7.Conclusions are not related to the title and purpose of the article. It is suggested that they be elaborated again and include general aspects and perspectives on these viral diseases in equines. For example, it is advisable to include the importance of the notification and circulation of these diseases, the need for homogeneity in diagnostic methods as well as the prevention of these diseases.

Minor points:

  1. The introduction is too long. The excessive comparison between horses and donkeys and mules is not relevant to the object of study. So is the evolutionary history and ancestors of these animals. It is recommended to summarize the aspects that are not related to the object of the work and highlight why these diseases are chosen to be reviewed.

2.It is recommended to increase Table 1 with other comparative aspects that are mentioned in the text, for example characteristics of blood, volume, and body weight.

Round 2

Reviewer 3 Report

The authors have answer all the concerns of this reviewer, only 2 minor details should be fixed

Simple summary line 21: . Implementation of proper biosecurity measures and vaccination programs, is totally without context.

Line 252 “The NPHV was identified in horses in 2012, and was considered the primary target of the virus [17]” Its not clear what the author mean, …and liver was considered..?
